# Nanoscale Zero-Valent Iron Modified by Bentonite with Enhanced Cr(VI) Removal Efficiency, Improved Mobility, and Reduced Toxicity

**DOI:** 10.3390/nano11102580

**Published:** 2021-09-30

**Authors:** Jien Ye, Yating Luo, Jiacong Sun, Jiyan Shi

**Affiliations:** 1Department of Environmental Engineering, College of Environmental and Resource Sciences, Zhejiang University, Hangzhou 310058, China; yejien@zju.edu.cn (J.Y.); luoyating@zju.edu.cn (Y.L.); wssjc111@126.com (J.S.); 2MOE Key Laboratory of Environment Remediation and Ecological Health, College of Environmental & Resource Science, Zhejiang University, Hangzhou 310058, China

**Keywords:** nanoscale zerovalent iron (nZVI), hexavalent chromium, bentonite, toxicity

## Abstract

The aggregation of nanoscale zero-valent iron (nZVI) particles and their limited transport ability in environmental media hinder their application in environmental remediation. In this study, the Cr(VI) removal efficiency, transport performance, and toxicity of nZVI and bentonite-modified nZVI (B-nZVI) were investigated. Compared with nZVI, B-nZVI improved the removal efficiency of Cr(VI) by 10%, and also significantly increased the transport in quartz sand and soil. Increasing the flow rate can enhance the transport of nZVI and B-nZVI in the quartz sand columns. The transport of the two materials in different soils was negatively correlated with the clay composition. Besides, modification of nZVI by bentonite could reduce toxicity to luminous bacteria (*Photobacterium phosphereum* T3) and ryegrass (*Lolium perenne* L.). Compared with Fe-EDTA, the transfer factors of nZVI and B-nZVI were 65.0% and 66.4% lower, respectively. This indicated that although iron nanoparticles accumulated in the roots of ryegrass, they were difficult to be transported to the shoots. The results of this study indicate that B-nZVI has a strong application potential in in situ environmental remediation.

## 1. Introduction

Nanoscale zerovalent iron (nZVI) is an engineered nanoparticle that is increasingly used in environmental remediation [1,2]. Due to its large specific surface area, strong adsorption, and reactivity to pollutants, nZVI has been reported to have a high removal efficiency on heavy metals, halogenated hydrocarbons, and other pollutants [3,4,5,6,7,8]. For example, nZVI showed great performance in the remediation of groundwater contaminated with hexavalent chromium [9]. The removal efficiency of pollutants is determined by a range of properties, including aggregation potential, particle surface properties, particle mobility, and chemical reactivity, and the characteristics of pollutants [10]. Due to its small size and magnetic properties, nZVI tends to agglomerate, which reduces its reactivity and mobility. Besides, nZVI can easily react with oxygen and/or water to form iron oxides layer on the particle surface, which prevents the further contact with pollutants [11]. Agglomeration and surface passivation are considered to be the challenges in the application of nZVI to environmental remediation [12]. In recent years, different modified methods have been reported to decrease the agglomeration and improve the mobility and reactivity of nZVI, such as the surface modification (chitosan [13,14], sodium carboxymethyl cellulose [15], starch [16], and polyethylene glycol [17]) and carrier modification (biochar [18], active carbon [19], silica [20], zeolite [21], attapulgite [22], and diatomite [23]). In addition, the modification of reaction conditions such as ultrasound and photocatalysis can also improve the removal efficiency of pollutants [24,25,26]. Bentonite is a widely used porous adsorbent, which has high adsorption capability, chemical and mechanical stability, and unique inter-lamellar structural properties [27]. Previous studies have proved that using bentonite as a support material for nZVI can improve the efficiency of Cr(VI) removal and enhance the mobility in porous media (quartz sand column) [11,28]. However, the transport performance of bentonite-modified nZVI in soil has not been reported.

In addition to agglomerate, the potential hazards of nZVI have also raised great concern [29]. As an efficient material that is increasingly used in site remediation, the potential ecological toxicity and risks are still largely unknown [30]. The application of nZVI induced the decrease of pH but the increase in redox potential in Zn-polluted soil [31]. Wang et al. reported that high concentration (>500 mg L^−1^) of nZVI inhibited the rice seedlings growth and inhibited active iron transportation in rice plants [32]. In the soil microbiota, the nZVI also has a significant effect which varies according to the type of soil [33,34,35]. At present, the toxicity mechanism of nZVI has not been fully understood, but the main possible mechanisms are the release of iron ions and oxidative stress. Once in the environment, nZVI is oxidized and continuously releases iron ions [36]. Iron ions are indispensable for the growth of organisms, but excessive iron ions will cause harm to organisms [37]. Moreover, due to the small size, large specific surface area, and abundant active sites for electron donors and acceptors on the particle surface, nZVI can easily react with molecular oxygen (O_2_) to form reactive oxygen species (ROS) [33]. If excessive reactive oxygen species cannot be eliminated by the antioxidant defense system in the cell, oxidative stress will occur and cause cell death [38]. Therefore, it is necessary to clarify the toxicity of nZVI and its modified materials.

The main aims of this study are: 1) To synthesize nZVI and bentonite modified nZVI (B-nZVI) and investigate the Cr(VI) removal efficiency of these two materials; 2) to investigate the mobility of nZVI and B-nZVI in quartz sand and soil columns; 3) to determine their toxicity toward bacteria and plants. Results of this study could provide a theoretical basis for the application of iron-based nanomaterials in environmental remediation.

## 2. Materials and Methods

### 2.1. Materials and Chemicals

Ferric chloride hexahydrate (FeCl_3_∙6H_2_O), bentonite (Al_2_O_3_·4(SiO_2_)·H_2_O), anhydrous ethanol, potassium borohydride (KBH_4_), polyvinylpyrrolidone (PVP, K29-32), and potassium dichromate (K_2_Cr_2_O_7_) were purchased from Aladdin Bio-Chem Technology Co., Ltd., Shanghai, China.

### 2.2. Preparation of nZVI and Bentonite-Supported nZVI

Nanoscale zero-valent iron (nZVI) and bentonite-supported nZVI (B-nZVI) were prepared through a modified chemical reduction [28,39]. Briefly, 1.35 g FeCl_3_∙6H_2_O, 0.28 g bentonite, and 0.5 g PVP (K29-32) were dissolved in 100 mL ethanol–water solution (30% V/V) in a three-necked flask. After mechanical stirring under N_2_ environment for 10 min, 100 mL 0.2 M KBH_4_ (dissolved in 1% NaOH) was added dropwise by a constant pressure separation funnel. Ferric ions were reduced to zero-valent iron by the following reaction:(1)$${4\mathrm{Fe}}^{3+}{+3\mathrm{BH}}_{4}^{-}{+9H}_{2}O\;\to {\;4\mathrm{Fe}}^{0}↓{+3H}_{2}{\mathrm{BO}}_{3}^{-}{+12H}^{+}{+6H}_{2}↑$$

After mechanical stirring under N_2_ environment for 30 min, the products were washed with anhydrous ethanol for three times. The products were dried under vacuum and sealed storage. The preparation of nZVI used the same procedure, but without bentonite.

### 2.3. Characterization of nZVI and B-nZVI

The morphology and elementary composition of nZVI and B-nZVI was characterized by scanning electron microscope (Gemini SEM 300, Carl Zeiss AG, Oberkochen, Germany) and transmission electron microscopy (JEM 1200EX, Hitachi, Tokyo, Japan). Gas absorption operation was carried out for the determination of particles’ specific surface area in a surface area analyzer (ASAP 2460, Micromeritics, Norcross, GA, USA). X-ray diffraction (XRD) was performed in a X-ray diffractometer (Ultima, Rigaku, Takatsuki, Japan) with Cu Kα radiation (λ = 0.154 nm). The scan range was set from 10° to 90° at 40 kV/40 mA. X-ray photoelectron spectroscopy was operated on a X-ray photoelectron spectrometer (K-Alpha, Thermo Fisher Scientific, Waltham, MA, USA) with Al Kα radiation (hv = 1486.6 eV) at 12 kV/6 mA.

### 2.4. Batch Experiments

A solution with 300 mg L^−1^ Cr(VI) was prepared by dissolving desired amount of K_2_Cr_2_O_7_, and the initial pH of the solution was adjusted by 0.2 M NaOH or HCl. Cr(VI) removal experiments were performed using conical flasks (500 mL) with an oscillation frequency of 200 rpm. The removal efficiency of Cr(VI) (300 mg L^−1^, 100 mL) by nZVI and B-nZVI (0.1 g, based on the mass of Fe) was investigated at different initial pH (4.0, 7.0, and 10.0). After 2, 5, 10, 20, 30, 60, 120, 240, 360, 960, and 1440 min of reaction, 2 mL solution was sampled and filtered through a 0.22 μm membrane filter. The concentrations of Cr(VI) in the solutions were determined using the diphenylcarbohydrazide method [40]. The absorbance of Cr(VI)-diphenylcarbohydrazide product was measured on a UV-vis spectrometer (UV-1800, Shimadzu, Kyoto, Japan).

### 2.5. Column Transport Experiments

The column used for column transport experiments was a cylindrical plexiglass column with a length of 10 cm and an inner diameter of 1.3 cm. The column transport experiments were carried out in two media: quartz sand (20–30 mesh) and soils (Appendix A). The concentration of nZVI and B-nZVI suspension in this experiment was 0.50 g L^−1^, and the background solution was NaCl solution prepared with anaerobic water (10 mM, pH 7.5). The column was first injected with four pore volumes (PVs) background solution in up-flow mode by a peristaltic pump to ensure that the column reaches water saturation and a stable flow state. Subsequently, 4 PV nZVI or B-nZVI suspension and 4 PV background solution were pumped into the column. The suspensions were continuously sonicated to avoid the sedimentation of nZVI and B-nZVI. The effluent was mixed with hydrochloric acid in a ratio of 1:1 with a water bath at 90 °C for 1 h. The total Fe content in the solution was determined by flame atomic absorption (Pinaacle 900F, PerkinElmer, Singapore City, Singapore).

### 2.6. Luminous Bacteria Toxicity Test

The luminous bacteria (*Photobacterium phosphereum* T3) was purchased from Institute of Soil Science, Chinese Academy of Sciences (Kuake Technology Ltd., Nanjing, China). The freeze-dried luminous bacteria were mixed with 1 mL 2% NaCl solution in an ice bath for 2 min. Subsequently, 10 μL of the luminous bacteria solution was added into 2 mL sample solution (prepared with 3% NaCl). After being incubated at 20 °C for 15 min, the bioluminescence intensity was measured by DXY-2 microtox test instrument (Kuake Technology Ltd., Nanjing, China). Each sample was determined at least three times. HgCl_2_ was used as a reference in this test, and the relative fluorescence intensity had a linear relationship with the concentration of HgCl_2_ (0 and 0.2 mg L^−1^ correspond to 100% and 0% relative bioluminescence intensity, respectively) [41]. The toxicity of control (non-materials treated, CK) and ionic treatments using Fe-EDTA (50, 200, and 500 mg L^−1^) was evaluated in addition to the nZVI and B-nZVI treatments.

### 2.7. Ryegrass Hydroponic Experiment

Selected uniform perennial ryegrass (*Lolium perenne* L.) seeds were surface sterilized in 3% hydrogen peroxide solution for 30 min and thoroughly washed with deionized water. Then the ryegrass seeds were transferred to moist gauze and incubated in darkness at 28 °C for 6 days. The 6-day-old seedlings were transferred to a hydroponic solution with 1/3 strength heavy elements (1 mM Ca(NO_3_)_2_, 0.5 mM Ca(H_2_PO_4_)_2_, 0.5 mM K_2_SO_4_, 1 mM MgSO_4_, and 1.5 mM NH_4_NO_3_) and full strength trace elements (75 μm EDTA-Fe, 46 μm H_3_BO_3_, 9 μm MnSO_4_, 0.8 μm ZnSO_4_, 0.3 μm CuSO_4_, and 0.8μm Na_2_MoO_4_). The hydroponic solution was adjusted to pH 5.8. Different doses of nZVI and B-nZVI (50, 200, 500 mg L^−1^) were added into the hydroponic solution for the treatment groups. Control (non-materials treated, CK) and ionic treatments using Fe-EDTA (50, 200, and 500 mg L^−1^) were prepared in addition to the nZVI and B-nZVI treatments; three parallels for each treatment, and 24 seedlings for each parallel. The plant culture was performed in a growth chamber at 25 °C for 16 h in the day and at 20 °C for 8 h in the night with 50–60% relative humidity. The hydroponic solution was replaced every 3 days.

After 21 days of growth, ryegrass was harvested and washed with deionized water. Then shoots and roots were separated, and their dry weights were measured. The contents of Fe in plants were determined by ICP-MS (Plasmaquant MS, Analytik Jena AG, Jena, Germany) after being digested with HNO_3_-HClO_4_ (4:1, V/V). Fresh plant samples were mixed with 4 °C phosphate buffer (PBS), homogenized in a glass homogenizer. The homogenate was centrifuged at 4 °C, and the supernatant was separated for the determination. SOD activity assay kit (S0109, Beyotime Biotechnology Ltd., Shanghai, China) and catalase assay kit (S0051, Beyotime Biotechnology Ltd., Shanghai, China) were used for enzyme activity assay. Lipid peroxidation in plants was determined by MDA content by 2-thiobarbituric acid (TBA) reaction (S0131S, Beyotime Biotechnology Ltd., Shanghai, China).

## 3. Results and Discussion

### 3.1. Characterization of nZVI and B-nZVI

The morphology and size of the newly prepared nZVI and B-nZVI were characterized by scanning electron microscopy (SEM) and transmission electron microscopy (TEM) (Figure 1). The SEM images showed that nZVI were basically spherical and aggregated in a chain-like structure, which can be mainly caused by the magnetic properties of the materials [42]. The size of each zero-valent iron particle was between 50 and 100 nm (Figure 1c). As for B-nZVI, the iron particles were loaded on the surface of bentonite, and the overall size of the material depended on the size of the supporter. The TEM and EDX analysis of the materials before and after the reaction showed that the morphology of the materials had no obvious change after the reaction (Appendix A), while further agglomeration of nZVI occurred. After the reaction, the appearance of chromium element indicated that part of the chromium was adsorbed on the surface of the particles (Appendix A). The Brunauer–Emmet–Teller (BET) isotherm was used to determine the specific surface area and pore size of nZVI and B-nZVI (Appendix A). The results showed that the modification of bentonite can increase the specific surface area of nZVI (12.8 m^2^/g) to 26.5 m^2^/g, and the adsorption/desorption average pore diameters of nZVI and B-nZVI were 16.6/6.1 nm and 16.2/8.4 nm, respectively. As observed from the XRD patterns of nZVI and B-nZVI (Appendix A), a diffraction peak at the 2*θ* of 44.8° occurred both in nZVI and B-nZVI, which indicated the presence of the (110) facet of bcc iron (JCPDS no. 06-0696) [43]. No other obvious characteristic peak can be observed in the XRD pattern of nZVI. After modification by bentonite, diffraction peaks at the 2*θ* of 19.8°, 35.8°, and 62.8° can correspond to montmorillonite (JCPDS no. 29-1498) [44], which indicated the successful modification of nZVI with bentonite. In addition, a diffraction peak at the 2*θ* of 27.8° was observed in the pattern of B-nZVI, which is consistent with the characteristic peak of silicon oxide (JCPDS no. 82-1566), one of the main components of bentonite.

### 3.2. Cr(VI) Removal by nZVI and B-nZVI

In order to investigate the Cr(VI) removal efficiency of nZVI and B-nZVI, the removal experiments of Cr(VI) were studied under different initial pH conditions (4, 7, and 10). The results showed that for both nZVI and B-nZVI, the removal process of Cr(VI) was very fast (Figure 2). The removal amount reached 90% of the maximum removal efficiency in the first 10 min, and the concentration of Cr(VI) in the solution remained basically unchanged after 30 min. Compared with nZVI, the removal efficiency of Cr(VI) by B-nZVI can be improved by about 10%. This may be due to that iron nanoparticles are distributed on the surface of bentonite after modification by bentonite, which reduces the aggregation of iron nanoparticles and decreases the hydrodynamic diameter of the material (Appendix A). It can be also found from the results of TEM that modification of bentonite can alleviate the agglomeration of nZVI (Appendix A). In addition, we investigated the adsorption capacity of bentonite for Cr(VI) under different initial pH (Appendix A). The results showed that although bentonite did have a certain adsorption capacity for Cr(VI), its adsorption capacity was much lower than the improvement of the removal capacity of Cr(VI) by the modified B-nZVI. The removal efficiency of the two materials for Cr(VI) decreased with the increase of pH. When the pH increased from 4 to 10, the removal efficiency of nZVI and B-nZVI for Cr(VI) decreased by 13.7% and 10.4%, respectively. The high removal efficiency of Cr(VI) by nZVI at acidic conditions has been reported in previous studies [33,45]. The reaction between Fe^0^ and Cr(VI) is a process of consuming hydrogen ions, and thus the increased concentration of hydrogen ions in the solution can promote the reaction to proceed to the positive direction [46]. In addition, when under acidic conditions, hydrogen ions in the solution can dissolve iron oxide film, and increase the active site on the surface of iron particles [40,47].

Further analysis of the reaction products by XPS showed that the reaction products were mainly composed of Fe, Cr, O, and C elements (Figure 3). The Cr 2p_3/2_ and Cr 2p_1/2_ photoelectron peaks at 576.8 eV and 586.5 eV represent that Cr(OH)_3_, Cr(OH)O, and Cr_2_O_3_ are the main species of Cr [48]. The spin-orbit splitting energy is 9.8 eV and full-width at half-maximum (FWHM) is 3.0 eV, which also proves that Cr(III) is the dominant form of Cr on the particle surface [13,49]. In addition, a small amount of Cr(VI) products were detected in the reaction products of B-nZVI (Figure 3b). The binding energy of is 579.9 eV, representing the existence of K_2_Cr_2_O_7_ [50]. This may be due to the fact that since Cr(VI) in the solution was not completely reduced after the reaction, the residual Cr(VI) can be adsorbed by bentonite, resulting in the detection of Cr(VI) in the separated reaction products. Analysis of the Fe 2p spectra (Appendix A) shows that the binding energies of Fe2p_3/2_ and Fe2p_1/2_ are 710.9 eV and 724.5 eV, respectively. This indicates that Fe in the product basically exists in the form of Fe(III), and the possible chemical structure is iron oxide hydrate (FeOH), magnetite (Fe_3_O_4_), and hematite (Fe_2_O_3_) [51]. In addition, a photoelectron peak of 706.8 eV was found in the spectra of nZVI, indicating the presence of Fe^0^ after the reaction, and no such photoelectron peak was found in the spectra of B-nZVI. Li et al. also found that Fe^0^ can be still detected in the nZVI after reacting with Cr(VI) for 24 h [52]. This can be explained that the modification of bentonite reduced the aggregation of nZVI and caused the oxidation of Fe^0^ more sufficient, thus Fe^0^ was completely oxidized to Fe(III) after the reaction.

### 3.3. Transport of nZVI and B-nZVI in Quartz Sand Columns

The transport behaviors of nZVI and B-nZVI were studied by column transport experiments. The breakthrough curves of two materials begin at 1 PV and reach their plateaus at about 1.5 PV (Figure 4). For nZVI, its mobility in quartz sand column is very low, only <0.05 when it reaches the plateaus (Figure 4a). This may be due to the aggregation of nZVI in solution, which leads to the increase of particle size and the decline of mobility of nanoparticles [53]. In addition, in this experiment, the zeta potential of nZVI is positive and its surface is positively charged, while the zeta potential of quartz sand is negative and its surface is negatively charged (Appendix A). Thus, there is electrostatic attraction between the nanoparticles and quartz sand and nZVI tends to adsorb on the surface of quartz sand [28]. Moreover, the mobility of nZVI in the quartz sand column is affected by the flow rate. With the flow rate increasing from 1.0 cm min^−1^ to 3.0 cm min^−1^, the plateaus increase from 0.02 to 0.04. Previous studies have also found that increasing the flow rate can improve the mobility of nanoparticles, mainly because increasing the flow rate can generate stronger hydrodynamic forces to keep the colloids in the mobile phase [54,55].

After modification with bentonite, the mobility of nZVI in quartz sand column has been greatly improved (Figure 4b). At the flow rate of 3.0 cm min^−1^, the breakthrough plateaus can reach 0.35, which is about eight times that of nZVI. The attachment of nZVI to bentonite decreased the magnetic attraction and created steric repulsion between nanoparticles, thus facilitating the transport of nZVI through quartz sand column [18]. Moreover, the modification also changed the surface charge of the materials. After modification with bentonite, the zeta potential of the material turns to a negative value (Appendix A). The electrostatic repulsion between the nanoparticles and quartz sand reduces the deposition of the material on the surface of the quartz sand [28]. Similar to nZVI, increasing the flow rate can also increase the mobility of B-nZVI in the quartz sand column. Besides, increasing the mass ratio of carrier was also found to facilitate the transport of B-nZVI in quartz sand column. Zhang et al. found that when the mass ratios of alginate and bentonite were greater than 0.5 and 10, respectively, the modified nZVI can completely penetrate the quartz sand column [28].

### 3.4. Transport of nZVI and B-nZVI in Soil Columns

The mobility of nZVI and B-nZVI in soil columns packing with three different soils was studied (Figure 5). Similar to the situation in the quartz sand column, the breakthrough curves of nZVI and B-nZVI start at 1 PV and reached their plateaus at about 1.5 PV. The iron concentration in the effluent increased continuously after the nZVI suspension was pumped into the soil column. The breakthrough curves of nZVI reached the plateau of < 0.04 at about 4 PV, which indicates that the mobility of nZVI in the soil column is low. The poor mobility of nZVI in soil columns has also been reported previously [56,57]. Dong et al. studied the transport behavior of nZVI in saturated sand-and soil-packed columns and found that surface modification affected the transport process [58]. Compared with nZVI, the modification with bentonite significantly improved its breakthrough curve plateaus in the three different soils, which increased by 7.4, 2.2, and 10.3 times in S1, S2, and S3, respectively. Soil texture has a great influence on the mobility of nZVI and B-nZVI, and the mobility of the two materials in the three soils are S2 > S1 > S3. The higher the sand content of the soil, the lower the clay content, and the better is the mobility of the material in the soil (Appendix A). Previous studies have shown that the soil adsorption capacity has a positive relation with the clay content, which can be ascribed to its large surface area and higher porosity [59,60].

### 3.5. Luminous Bacteria Toxicity Test

The toxicity of nZVI and B-nZVI was evaluated by luminous bacteria toxicity test (Figure 6). The relative bioluminescence intensity of luminous bacteria was negatively correlated with the toxicity of the material [41]. The results showed that both nZVI and B-nZVI were toxic to luminous bacteria, and the toxicity increased obviously with the increase of the concentration of materials. The addition of 500 mg L^−1^ of nZVI can reduce the relative bioluminescence intensity by 93.7%, which indicates that high concentration of nZVI is highly toxic to the luminous bacteria. Some studies reported that nZVI had no cytotoxicity toward microorganisms. For example, 1000–10,000 mg/L of nZVI had no effect on the survival and activity of *Klebsiella planticola*, and did not cause significant cell damage [61]. However, more studies reported that nZVI had a significant negative impact on the survival rate and biological activity of microorganisms. Studies have found that more than 70% of *Escherichia coli* were inactivated with the treatment of 70 mg/L nZVI [62]. Compared with *Escherichia coli*, *Bacillus subtilis* showed a stronger tolerance to nZVI, but the survival rate of *Bacillus subtilis* also decreased with the extension of nZVI treatment time [63]. Surface modification of nZVI also affects its cytotoxicity [64]. After the modification by bentonite, the relative bioluminescence intensity had no significant changes with 50 and 200 mg L^−1^ treatments. However, the relative bioluminescence increased from 6.3% to 14.6% with the treatment of 500 mg L^−1^, indicating that the toxicity of B-nZVI was lower than nZVI. In the study reported by Qiu et al. [65], the presence of 150 mg L^−1^ nZVI can decrease the relative bioluminescence intensity of luminous bacteria by 49.3%, and the generated iron ions were considered to be the main source of toxicity. In order to figure out the effect of iron ions to luminous bacteria, Fe-EDTA treatments were prepared as ionic controls. Compared with nZVI and B-nZVI, the treatment of 50 mg L^−1^ Fe-EDTA had no significant effect on the relative bioluminescence intensity. The toxicity of Fe-EDTA to luminous bacteria was produced as the treated concentration was increased. However, compared with the same concentration of nanomaterials, its toxicity was significantly reduced. This suggests that the toxicity of nZVI and B-nZVI to luminous bacteria may not only come from the dissolved iron ions, but also from the nanoparticles themselves. Metal-based NPs have been found to induce perturbations on microbes not only by the dissolved ions, but also by the particles themselves [66]. Due to the high specific surface area and numerous active sites on the surface, nZVI can react with oxygen to form reactive oxygen species (ROS) such as ·OH, O_2_^−^, and H_2_O_2_ [67]. The excessive accumulation of ROS will denature macromolecules including lipids, proteins, and nucleic acids, destroy the cell structure, and eventually lead to cell death [68].

### 3.6. Ryegrass Hydroponic Experiment

The phytotoxicity of nZVI and B-nZVI was investigated by hydroponic experiment of ryegrass. The results indicated that the addition of 50 and 200 mg L^−1^ of nanomaterials had no significant effects on the biomass of the plants (Appendix A). However, there was a significant decline in biomass of the shoots (approximately 50%) and roots (approximately 20–30%) with the treatment of 500 mg L^−1^ nZVI and B-nZVI. Compared with nanomaterials, none of the Fe-EDTA treatments affected the biomass of the ryegrass shoots. Only the 500 mg L^−1^ Fe-EDTA treatments inhibited the biomass of the ryegrass shoots (approximately 30%) after 21 days of hydroponic culture. Although iron is one of the essential nutrients for plants, excessive iron concentration can inhibit plant growth [37].

In order to identify the bioavailability of nZVI and B-nZVI, Fe accumulation in plants was further analyzed (Appendix A). The results showed that the total Fe content in the roots of ryegrass treated with nZVI and B-nZVI were much higher than that in the control group, and the Fe content with the treatment of 500 mg L^−1^ nanomaterials was 10.5 times and 5.6 times that of the control group, respectively. The Fe accumulation in the shoots also increased by 2–3 times compared with the control group, while the Fe content in the group treated with B-nZVI was relatively independent of the dosage of B-nZVI. Similarly, the total Fe content in the plants with the treatment of Fe-EDTA was higher than in the control group. However, compared with the same concentration of nanomaterials treatment groups, the Fe-EDTA treatment group had a lower Fe content in the roots. As a result, the translocation factor (C_shoot_/C_root_) of the Fe-EDTA treatment group was higher than that of the nanomaterials-treated groups (Appendix A). In other words, nanomaterials were more likely to accumulate in the roots of ryegrass rather than being transported to the shoots. Previous studies have reported similar results that metal-based nanoparticles taken up by the roots were difficult to transport to the aerial part of the plants [69,70]. Wang et al. found that the addition of high concentration of nZVI decreased the effective iron content in rice plants [32]. The iron deficiency was not caused by deficiency of available iron in the soil or restraint of the absorption that plant takes in the available iron, but the damage of the root cortex tissues. As a result, the transport of available iron from roots to shoots was blocked.

An increasing number of research have reported that iron-based NPs can cause oxidative stress and lipid peroxidation to plants [71,72,73,74]. The activities of antioxidative enzymes (SOD, CAT) and the content of MDA in the shoots and roots of ryegrass are shown in Figure 7. The results showed that the treatments of nZVI, B-nZVI, and Fe-EDTA had no significant effect on SOD activity in the shoots of ryegrass. SOD activity in the ryegrass roots increased by 78.1% and 65.5% with the treatments of 500 mg L^−1^ nZVI and B-nZVI, respectively (p < 0.05, Figure 7a). The addition of 500 mg L^−1^ Fe-EDTA also increased the activity of SOD in the roots, but it was lower than that of the iron-based NPs groups. CAT activities were also enhanced by 63.8–82.3% with the treatments of 200 and 500 mg L^−1^ nZVI and B-nZVI (Figure 7b). These data suggest that high concentration of both the iron-based NPs and Fe-EDTA treatments can cause oxidative stress. Oxidative stress can be further evidenced by the content of MDA. Compared with the control group, the MDA contents increased significantly in the roots with the treatments of 500 mg L^−1^ nZVI and B-nZVI. Excessive ROS produced in high-concentration treatments can induce lipid peroxidation, which change the fluidity and permeability of cell membranes, and ultimately lead to changes in cell structure and function [67]. Due to its high reactivity, nZVI can denature lipopolysaccharide and ion membrane transporter on the cell membrane and reduce the permeability of the cell membrane, thus Fe^2+^ ions produced by nZVI can enter the cell. Fe^2+^ ions can react with H_2_O_2_ produced by mitochondria by Fenton-like reaction to produce ROS [75], which will eventually cause oxidative stress and cell death.

Overall, the treatments of iron-based NPs had greater effects on the roots of plants rather than the shoots. The results of antioxidative enzymes and lipid peroxidation correlated well with the results of biomass and Fe contents. Excessive iron accumulation can inhibit plants growth, cause oxidative stress, and stimulate plants to up-regulate antioxidative enzymes activity. Oxidative stress caused by iron-based NPs has been reported in other research. The exposure of 100 mg L^−1^ Fe_3_O_4_ NPs significantly induced oxidative stress and posed toxicity risks to the ryegrass and pumpkin plants [76]. High dose (500 mg L^−1^) of nZVI has also been found to up-regulate the CAT activity and MDA content in rice plants [77].

## 4. Conclusions

In this study, it was found that modification of nZVI by bentonite could improve the removal efficiency of Cr(VI) by 10%. The transport of nZVI was limited both in quartz sand and soil columns, and the transport performance can be improved by increasing the flow rate. Compared with nZVI, the transport of B-nZVI in quartz sand and different types of soil was increased by 8 times and 2–10 times, respectively. The transport of the two materials in different soils was positively correlated with the sand composition. Modification of nZVI by bentonite could reduce the toxicity to luminous bacteria (*Photobacterium phosphereum* T3) and perennial ryegrass (*Lolium perenne* L.). The transfer factors in nZVI and B-nZVI treatments were much lower than those in the Fe-EDTA treatments. This indicated that although iron nanoparticles accumulated in the roots of ryegrass, they were difficult to be transported to the shoots. The results show that bentonite is an ideal carrier, which can not only enhance the removal ability of nZVI to Cr(VI) and improve its migration performance in porous media, but also reduce its ecological toxicity. B-nZVI has a strong potential in the remediation of groundwater and soil polluted by Cr(VI).

## Figures and Tables

**Figure 1 nanomaterials-11-02580-f001:**
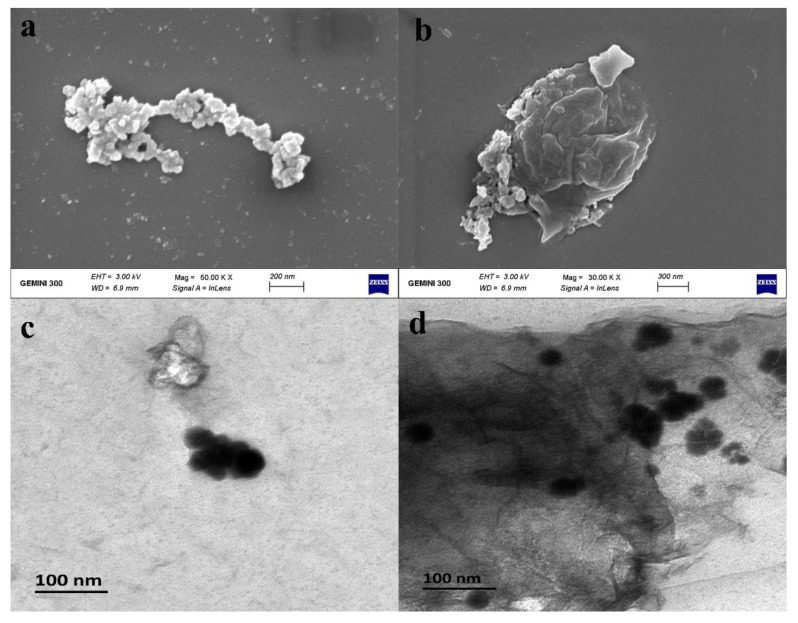
SEM and TEM images of nZVI (**a**,**c**) and B-nZVI (**b**,**d**).

**Figure 2 nanomaterials-11-02580-f002:**
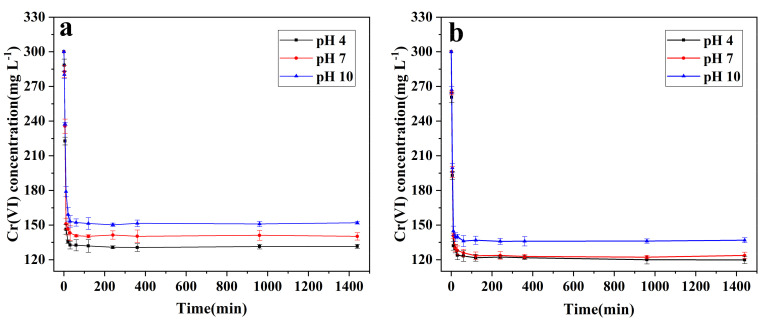
Effect of initial pH on Cr(VI) removal by nZVI (**a**) and B-nZVI (**b**). Error bars indicate the standard deviation of the mean (n = 3).

**Figure 3 nanomaterials-11-02580-f003:**
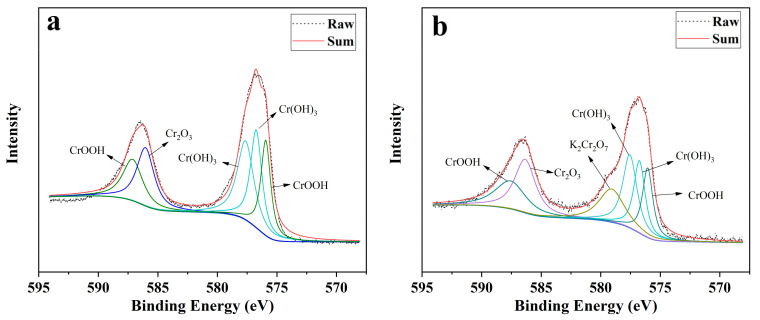
Cr 2p XPS spectra of reacted nZVI (**a**) and B-nZVI (**b**).

**Figure 4 nanomaterials-11-02580-f004:**
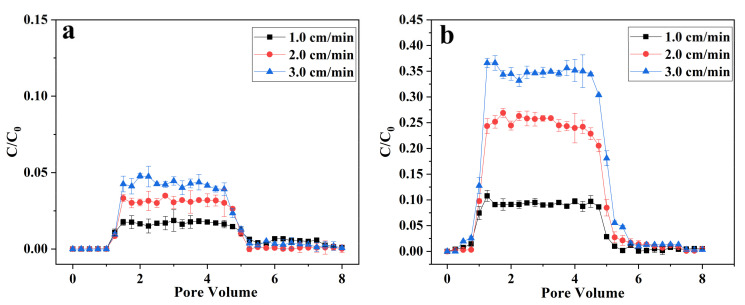
Breakthrough curves of nZVI (**a**) and B-nZVI (**b**) at different pore velocities through quartz sand columns. Error bars represent standard deviations of triplicate experiments.

**Figure 5 nanomaterials-11-02580-f005:**
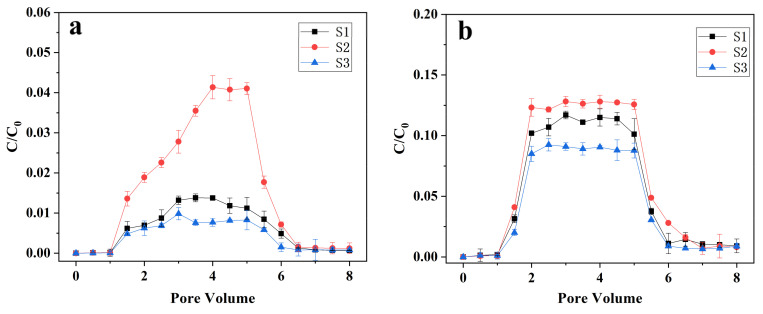
Breakthrough curves of nZVI (**a**) and B-nZVI (**b**) through columns packed with different soils. Error bars represent standard deviations of triplicate experiments.

**Figure 6 nanomaterials-11-02580-f006:**
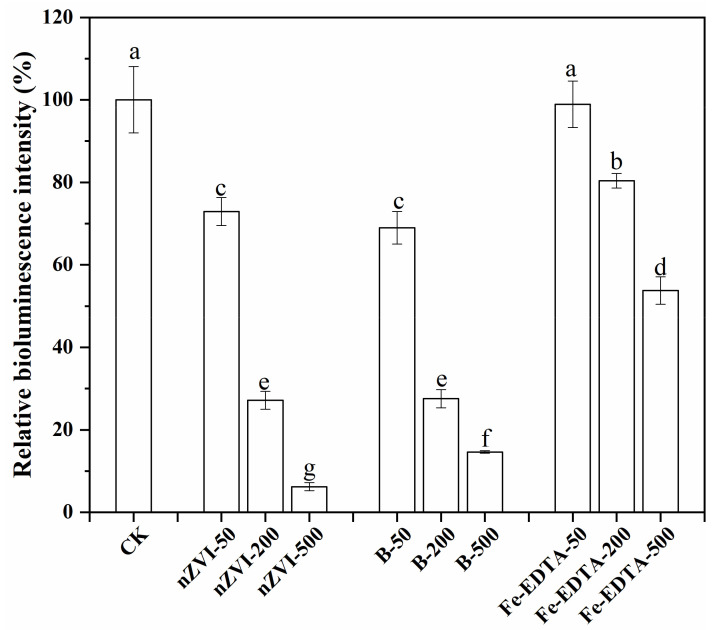
The relative bioluminescence intensity of different treatment groups. Error bars indicate the standard deviation of the mean (n = 3). Lowercase letters (a - f) represent significant differences among the different treatments determined by the Fisher least significant difference (LSD) test (*p* < 0.05).

**Figure 7 nanomaterials-11-02580-f007:**
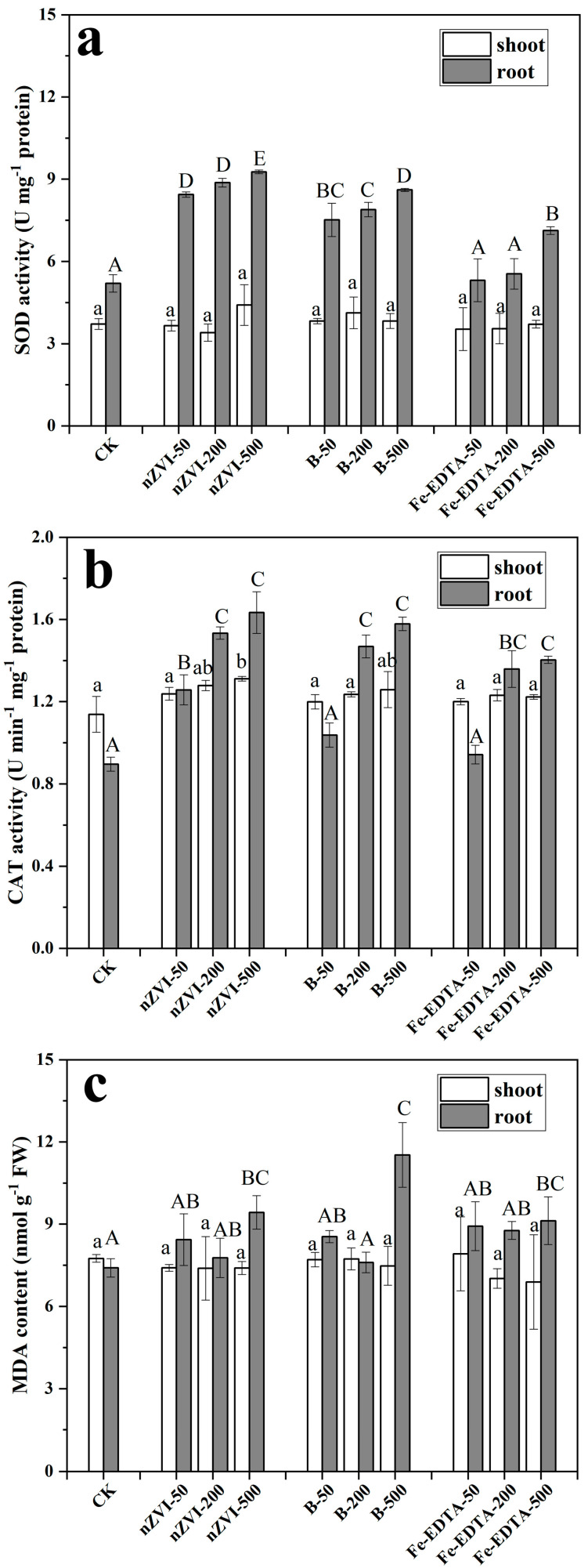
SOD (**a**), CAT (**b**) activities and MDA (**c**) contents in perennial ryegrass. Error bars indicate the standard deviation of the mean (n = 3). Lowercase and uppercase letters represent significant differences of shoots and roots among the different treatments determined by the Fisher least significant difference (LSD) test (*p* < 0.05).

## Data Availability

Data are contained within the article and Appendix A.

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
