# Peer review of "Nanoscale Zero-Valent Iron Modified by Bentonite with Enhanced Cr(VI) Removal Efficiency, Improved Mobility, and Reduced Toxicity"

_nanomaterials, 2021, doi:10.3390/nano11102580_

Round 1

Reviewer 1 Report

The main aims of this study were: 1) to synthesize nZVI and bentonite modified nZVI (B-nZVI) and investigate the Cr(VI) removal efficiency of these two materials; 2) to investigate the mobility of nZVI and B-nZVI in quartz sand and soil coloums; 3) to determine their toxicity towards bacteria and plants. The paper contains original (unpublished before) information. I found the manuscript very superficial in the explanations of the main processes in act. These results may be interesting to some of Nanomaterials readers. However, revision is required before accepting for publication.

Page 3 line 109: “reach” should be replaced with “reaches”.

Page 3 line 109: “stable flow state” should be replaced with “a stable flow state”.

Page 3 line 111: “was pumped” should be replaced with “were pumped”.

Page 3 line 143: “dry weight” should be replaced with “dry weights”.

Page 4 line 164: “be corresponded” should be replaced with “can correspond”.

Page 5 line 182: “certain adsorption” should be replaced with “a certain adsorption”.

Page 5 line 188: “…The reaction between of Fe0 and Cr(VI)…” should be replaced with “…The reaction between Fe0 and Cr(VI)…”.

Page 8 line 293: “…on biomass…” should be replaced with “…in biomass…”.

Page 11 line 350: “…lower than that in the Fe-EDTA treatments” should be replaced with “lower than those in the Fe-EDTA treatments”.

The authors should discuss in more detail the mechanism of the process.

The introduction part should be rewritten by including new references. Following refs. can be included in the introduction part.

Separation and Purification Technology (2021) 277, 119488

ACS Applied Materials & Interfaces (2021) 13 (11), 13072

Ultrasonics Sonochemistry (2020) 64, 105044

International journal of environmental research and public health (2021) 18 (11), 5921

International Journal of Environmental Research and Public Health (2021) 18 (11), 5887

Chemosphere (2021) 267, 129183

Results and discussion has been poorly written and should be strictly revised. It should be presented with clarity and precision and should be explained by referring to the literature and should interpret the findings in view of the results obtained.

N2 adsorption-desorption analysis should be performed to determine the specific surface area and pore size distribution of the samples.

It is suggested to investigate the stability of nZVI and B-nZVI by performing EDX analysis before and after the process.

Author Response

Replies to the comments of reviewer 1

General Comments

The main aims of this study were: 1) to synthesize nZVI and bentonite modified nZVI (B-nZVI) and investigate the Cr(VI) removal efficiency of these two materials; 2) to investigate the mobility of nZVI and B-nZVI in quartz sand and soil coloums; 3) to determine their toxicity towards bacteria and plants. The paper contains original (unpublished before) information. I found the manuscript very superficial in the explanations of the main processes in act. These results may be interesting to some of Nanomaterials readers. However, revision is required before accepting for publication.

Response: We gratefully acknowledge this reviewer for his or her favorable opinion and valuable comments to the paper. The manuscript has been revised according to the reviewer’s suggestions, and the point-by-point replies have been made as follow.

Specific Comments

Comment 1: Page 3 line 109: “reach” should be replaced with “reaches”; “stable flow state” should be replaced with “a stable flow state”. Page 3 line 111: “was pumped” should be replaced with “were pumped”. Page 3 line 143: “dry weight” should be replaced with “dry weights”. Page 4 line 164: “be corresponded” should be replaced with “can correspond”. Page 5 line 182: “certain adsorption” should be replaced with “a certain adsorption”. Page 5 line 188: “…The reaction between of Fe0 and Cr(VI)…” should be replaced with “…The reaction between Fe0 and Cr(VI)…”. Page 8 line 293: “…on biomass…” should be replaced with “…in biomass…”.Page 11 line 350: “…lower than that in the Fe-EDTA treatments” should be replaced with “lower than those in the Fe-EDTA treatments”.

Response: We appreciate the reviewer’s valuable comment. We have corrected these grammatical errors in the revised manuscript.

Comment 2: The authors should discuss in more detail the mechanism of the process.

Response: Thanks for the reviewer’s valuable comment. We have conducted a more in-depth discussion on the mechanism of Cr(VI) removal, transport and toxicity of nZVI and B-nZVI. Please see Results and Discussion in the revised manuscript.

Comment 3: The introduction part should be rewritten by including new references.

Response: We are very grateful for the reviewer’s comment and agree with it. We have rewritten the introduction part by including new references. Please see Page 1 line 34-37, line 39-42 and Page 2 line 46-48 in the revised manuscript.

Comment 4: Results and discussion has been poorly written and should be strictly revised. It should be presented with clarity and precision and should be explained by referring to the literature and should interpret the findings in view of the results obtained.

Response: Thanks for the reviewer’s valuable comment. We have supplemented the results and discussion section. Combined with the results of other studies, the mechanism in the process was further discussed. Please see Results and Discussion in the revised manuscript.

Comment 5: N2 adsorption-desorption analysis should be performed to determine the specific surface area and pore size distribution of the samples.

Response: Thanks for the reviewer’s valuable comment. We determined the specific surface area and pore size of nZVI and B-nZVI, and found that the modification of bentonite can increase the specific surface area of nZVI, but the pore size did not change significantly. The corresponding results have also been supplemented in the results and discussion (please see Page 4 line 178-183 in the revised manuscript and Figure S2 in the Supporting Information).

Comment 6: It is suggested to investigate the stability of nZVI and B-nZVI by performing EDX analysis before and after the process.

Response: Thanks for the reviewer’s valuable comment. We used TEM and EDX to analyze the materials before and after the reaction, and found that the elemental composition of the materials did not change significantly, but the agglomeration of nZVI increased after the reaction, and chromium was adsorbed on the materials. The corresponding results have also been supplemented in the results and discussion (please see Page 4 line 174-178 in the revised manuscript and Figure S1 in the Supporting Information).

Summary of revisions to second vision of manuscript

Original paper

revision

Revised paper

Page

Number

Line

Number

Page

Number

Line

Number

1

23-24

The sentence has been rewritten.

1

24-25

1

27-44

This paragraph has been rewritten.

1-2

29-53

2

52

A reference was added.

2

61

2

85-90

The characterization of TEM and BET was added.

3

96-99

3

109

“reach” was changed to “reaches”.

3

122

3

109

“stable flow state” was changed to “a stable flow state”.

3

122

3

111

“was pumped” was changed to “were pumped”.

3

124

3

143

“weight” was changed to “weights”.

4

157

4

169-170

Figure 1 was revised.

5

192-193

5

180-181

The result of TEM was added.

5

204-205

5

164

“certain” was changed to “a certain”.

5

207

5

188

“The reaction between of” was changed to “The reaction between”.

5

213

5

194

The caption of Figure 2 was revised.

6

219

5

211

A reference was added.

6

236-237

6

231-238

This paragraph has been rewritten.

7

258-272

7

250

A reference was added.

8

284-285

8

268

The discussion was supplemented.

8

304-312

8

283

The discussion was supplemented.

9

327-331

8

293

“on biomass” was changed to “in biomass”.

9

340

9

312

The discussion was supplemented.

10

360-365

9

326

The discussion was supplemented.

10

379-385

-

References were added.

12-13

453-458

13

465-469

13

481-482

13

486-494

13

512-514

14

532-533

14

549-551

14

570-572

15

578-585

15

590-593

Reviewer 2 Report

Comments

  1. It is well known that bentonite is impermeable. Did Bentonite presence cause some difficulties in the column experiments?
  2. English should be improved in some points e.g. the last sentence of Abstract is not well written.
  3. 3.1.and Supplementary material, XRD. Bentonite is a rock not a mineral. Therefore there are no peaks of bentonite.
  4. 3.1.and Supplementary material, XRD. According to my previous comment the authors should find and state the main mineral in bentonite. Is it montmorillonite? Saponite? Other?
  5. 3.1. SEM. SEM magnification is not high enough to determine the size of nanoparticles the authors should add TEM images.
  6. There are some format errors that the authors should correct e.g. Figure 2.

Author Response

Replies to the comments of reviewer 2

Specific Comments

Comment 1: It is well known that bentonite is impermeable. Did Bentonite presence cause some difficulties in the column experiments?

Response: Thanks for the reviewer’s valuable comment. Bentonite is impermeable and is suspended in water. In our experiments, we found that when the surface of bentonite was loaded with nZVI, it would gradually settle in water like nZVI suspension, but the sedimentation rate is less than nZVI. In the column experiment, the suspensions were continuously sonicated to avoid the sedimentation of nZVI and B-nZVI. Our research results found that bentonite did not hinder the transport of nZVI in saturated porous media, but instead enhanced its mobility.

Comment 2: English should be improved in some points e.g. the last sentence of Abstract is not well written.

Response: Thanks for the reviewer’s valuable comment. We have corrected some grammatical errors and rewritten some sentences in the revised manuscript.

Comment 3: 3.1.and Supplementary material, XRD. Bentonite is a rock not a mineral. Therefore there are no peaks of bentonite. The authors should find and state the main mineral in bentonite. Is it montmorillonite? Saponite? Other?

Response: Thanks for the reviewer’s valuable comment. We re-analyzed the XRD results and found that the main substances of B-nZVI were montmorillonite and silica, both of which are the main components of bentonite. We have modified the description in the results and discussion and Figure S3 in the Supporting Information.

Comment 4: 3.1. SEM. SEM magnification is not high enough to determine the size of nanoparticles the authors should add TEM images.

Response: Thanks for the reviewer’s valuable comment. We analyzed the morphology of nZVI and B-nZVI before and after the reaction by TEM and the corresponding content has been supplemented in the revised manuscript (please see Results and discussion. line 174-178) and Supporting Information (please see Figure S1).

Comment 5: There are some format errors that the authors should correct e.g. Figure 2.

Response: Thanks to the reviewer for reminding us. We have modified the caption of Figure 2.

Summary of revisions to second vision of manuscript

Original paper

revision

Revised paper

Page

Number

Line

Number

Page

Number

Line

Number

1

23-24

The sentence has been rewritten.

1

24-25

1

27-44

This paragraph has been rewritten.

1-2

29-53

2

52

A reference was added.

2

61

2

85-90

The characterization of TEM and BET was added.

3

96-99

3

109

“reach” was changed to “reaches”.

3

122

3

109

“stable flow state” was changed to “a stable flow state”.

3

122

3

111

“was pumped” was changed to “were pumped”.

3

124

3

143

“weight” was changed to “weights”.

4

157

4

169-170

Figure 1 was revised.

5

192-193

5

180-181

The result of TEM was added.

5

204-205

5

164

“certain” was changed to “a certain”.

5

207

5

188

“The reaction between of” was changed to “The reaction between”.

5

213

5

194

The caption of Figure 2 was revised.

6

219

5

211

A reference was added.

6

236-237

6

231-238

This paragraph has been rewritten.

7

258-272

7

250

A reference was added.

8

284-285

8

268

The discussion was supplemented.

8

304-312

8

283

The discussion was supplemented.

9

327-331

8

293

“on biomass” was changed to “in biomass”.

9

340

9

312

The discussion was supplemented.

10

360-365

9

326

The discussion was supplemented.

10

379-385

-

References were added.

12-13

453-458

13

465-469

13

481-482

13

486-494

13

512-514

14

532-533

14

549-551

14

570-572

15

578-585

15

590-593

Round 2

Reviewer 1 Report

The authors have satisfactorily addressed my comments and I recommend the manuscript for publication.

Reviewer 2 Report

The authors have addressed the points raised in my previous review and publication of the manuscript as is recommended.

Round 3

Reviewer 2 Report

I believe that the manuscript can now be accepted.